# Unsupervised Transformation Learning via Convex Relaxations

**Tatsunori B. Hashimoto**     **John C. Duchi**     **Percy Liang**
Stanford University
Stanford, CA 94305
{thashim,jduchi,pliang}@cs.stanford.edu

## Abstract

Our goal is to extract meaningful transformations from raw images, such as varying the thickness of lines in handwriting or the lighting in a portrait. We propose an unsupervised approach to learn such transformations by attempting to reconstruct an image from a linear combination of transformations of its nearest neighbors. On handwritten digits and celebrity portraits, we show that even with linear transformations, our method generates visually high-quality modified images. Moreover, since our method is semiparametric and does not model the data distribution, the learned transformations extrapolate off the training data and can be applied to new types of images.

## 1   Introduction

Transformations (e.g, rotating or varying the thickness of a handwritten digit) capture important invariances in data, which can be useful for dimensionality reduction [7], improving generative models through data augmentation [2], and removing nuisance variables in discriminative tasks [3]. However, current methods for learning transformations have two limitations. First, they rely on explicit transformation pairs—for example, given pairs of image patches undergoing rotation [12]. Second, improvements in transformation learning have focused on problems with known transformation classes, such as orthogonal or rotational groups [3, 4], while algorithms for general transformations require solving a difficult, nonconvex objective [12].

To tackle the above challenges, we propose a semiparametric approach for unsupervised transformation learning. Specifically, given data points $x_1, \ldots, x_n$, we find $K$ linear transformations $A_1 \ldots A_K$ such that the vector from each $x_i$ to its nearest neighbor lies near the span of $A_1 x_i \ldots A_K x_i$. The idea of using nearest neighbors for unsupervised learning has been explored in manifold learning [1, 7], but unlike these approaches and more recent work on representation learning [2, 13], we do not seek to model the full data distribution. Thus, even with relatively few parameters, the transformations we learn naturally extrapolate off the training distribution and can be applied to novel types of points (e.g., new types of images).

Our contribution is to express transformation matrices as a sum of rank-one matrices based on samples of the data. This new objective is convex, thus avoiding local minima (which we show to be a problem in practice), scales to real-world problems beyond the $10 \times 10$ image patches considered in past work, and allows us to derive disentangled transformations through a trace norm penalty.

Empirically, we show our method is fast and effective at recovering known disentangled transformations, improving on past baseline methods based on gradient descent and expectation maximization [11]. On the handwritten digits (MNIST) and celebrity faces (CelebA) datasets, our method finds interpretable and disentangled transformations—for handwritten digits, the thickness of lines and the size of loops in digits such as 0 and 9; and for celebrity faces, the degree of a smile.

## 2 Problem statement

Given a data point $x \in \mathbb{R}^d$ (e.g., an image) and strength scalar $t \in \mathbb{R}$, a *transformation* is a smooth function $f : \mathbb{R}^d \times \mathbb{R} \to \mathbb{R}^d$. For example, $f(x, t)$ may be a rotated image. For a collection $\{f_k\}_{k=1}^K$ of transformations, we consider *entangled transformations*, defined for a vector of strengths $t \in \mathbb{R}^K$ by $f(x, t) := \sum_{k=1}^K f_k(x, t_k)$. We consider the problem of estimating a collection of transformations $f^* := \sum_{k=1}^K f_k^*$ given random observations as follows: let $p_X$ be a distribution on points $x$ and $p_T$ on transformation strength vectors $t \in \mathbb{R}^K$, where the components $t_k$ are independent under $p_T$. Then for $\tilde{x}_i \overset{\text{iid}}{\sim} p_X$ and $t_i \overset{\text{iid}}{\sim} p_T$, $i = 1, \dots, n$, we observe the transformations $x_i = f^*(\tilde{x}_i, t_i)$, while $\tilde{x}_i$ and $t_i$ are unobserved. Our goal is to estimate the $K$ functions $f_1^*, \dots, f_K^*$.

### 2.1 Learning transformations based on matrix Lie groups

In this paper, we consider the subset of generic transformations defined via matrix Lie groups. These are natural as they map $\mathbb{R}^d \to \mathbb{R}^d$ and form a family of invertible transformations that we can parameterize by an exponential map. We begin by giving a simple example (rotation of points in two dimensions) and using this to establish the idea of the exponential map and its linear approximation. We then use these linear approximations for transformation learning.

A *matrix Lie group* is a set of invertible matrices closed under multiplication and inversion. In the example of rotation in two dimensions, the set of all rotations is parameterized by the angle $\theta$, and any rotation by $\theta$ has representation $R_\theta = \begin{bmatrix} \cos(\theta) & -\sin(\theta) \\ \sin(\theta) & \cos(\theta) \end{bmatrix}$. The set of rotation matrices form a Lie group, as $R_\theta R_{-\theta} = I$ and the rotations are closed under composition.

**Linear approximation.** In our context, the important property of matrix Lie groups is that for transformations near the identity, they have local linear approximations (tangent spaces, the associated *Lie algebra*), and these local linearizations map back into the Lie group via the exponential map [9]. As a simple example, consider the rotation $R_\theta$, which satisfies $R_\theta = I + \theta A + O(\theta^2)$, where $A = \begin{bmatrix} 0 & -1 \\ 1 & 0 \end{bmatrix}$, and $R_\theta = \exp(\theta A)$ for all $\theta$ (here $\exp$ is the matrix exponential). The infinitesimal structure of Lie groups means that such relationships hold more generally through the exponential map: for any matrix Lie group $G \subset \mathbb{R}^{d \times d}$, there exists $\varepsilon > 0$ such that for all $R \in G$ with $\|R - I\| \leq \varepsilon$, there is an $A \in \mathbb{R}^{d \times d}$ such that $R = \exp(A) = I + \sum_{m \geq 1} A^m/m!$. In the case that $G$ is a *one-dimensional* Lie group, we have more: for each $R$ near $I$, there is a $t \in \mathbb{R}$ satisfying

$$R = \exp(tA) = I + \sum_{m=1}^{\infty} \frac{t^m A^m}{m!}.$$

The matrix $tA = \log R$ in the exponential map is the derivative of our transformation (as $A \approx (R - I)/t$ for $R - I$ small) and is analogous to locally linear neighborhoods in manifold learning [10]. The exponential map states that for transformations close to the identity, a linear approximation is accurate.

For any matrix $A$, we can also generate a collection of associated 1-dimensional manifolds as follows: letting $x \in \mathbb{R}^d$, the set $M_x = \{\exp(tA)x \mid t \in \mathbb{R}\}$ is a manifold containing $x$. Given two nearby points $x_t = \exp(tA)x$ and $x_s = \exp(sA)x$, the local linearity of the exponential map shows that

$$x_t = \exp((t - s)A)x_s = x_s + (t - s)Ax_s + O((t - s)^2) \approx x_s + (t - s)Ax_s. \tag{1}$$

**Single transformation learning.** The approximation (1) suggests a learning algorithm for finding a transformation from points on a one-dimensional manifold $M$: given points $x_1, \dots, x_n$ sampled from $M$, pair each point $x_i$ with its nearest neighbor $\overline{x}_i$. Then we attempt to learn a transformation matrix $A$ satisfying $\overline{x}_i \approx x_i + t_i A x_i$ for some small $t_i$ for each of these nearest neighbor pairs. As nearest neighbor distances $\|\overline{x}_i - x_i\| \to 0$ as $n \to \infty$ [6], the linear approximation (1) eventually holds. For a one-dimensional manifold and transformation, we could then solve the problem

$$\underset{\{t_i\}, A}{\text{minimize}} \sum_{i=1}^n \|t_i A x_i - (\overline{x}_i - x_i)\|_2. \tag{2}$$

If instead of using nearest neighbors, the pairs $(x_i, \overline{x}_i)$ were given directly as supervision, then this objective would be a form of first-order matrix Lie group learning [12].

**Sampling and extrapolation.** The learning problem (2) is semiparametric: our goal is to learn a transformation matrix $A$ while considering the density of points $x$ as a nonparametric nuisance variable. By focusing on the modeling differences between nearby $(x, \overline{x})$ pairs, we avoid having to specify the density of $x$, which results in two advantages: first, the parametric nature of the model means that the transformations $A$ are defined beyond the support of the training data; and second, by not modeling the full density of $x$, we can learn the transformation $A$ even when the data comes from highly non-smooth distributions with arbitrary cluster structure.

## 3 Convex learning of transformations

The problem (2) makes sense only for one-dimensional manifolds without superposition of transformations, so we now extend the ideas (using the exponential map and its linear approximation) to a full matrix Lie group learning problem. We shall derive a natural objective function for this problem and provide a few theoretical results about it.

### 3.1 Problem setup

As real-world data contains multiple degrees of freedom, we learn several one-dimensional transformations, giving us the following multiple Lie group learning problem:

**Definition 3.1.** *Given data $x_1 \ldots x_n \in \mathbb{R}^d$ with $\overline{x}_i \in \mathbb{R}^d$ as the nearest neighbor of $x_i$, the nonconvex transformation learning problem objective is*

$$\underset{t \in \mathbb{R}^{d \times K}, A \in \mathbb{R}^{d \times d}}{\text{minimize}} \sum_{i=1}^{n} \left\| \sum_{k=1}^{K} t_{ik} A_k x_i - (\overline{x}_i - x_i) \right\|_2. \tag{3}$$

This problem is nonconvex, and prior authors have commented on the difficulty of optimizing similar objectives [11, 14]. To avoid this difficulty, we will construct a convex relaxation. Define a matrix $Z \in \mathbb{R}^{n \times d^2}$, where row $Z_i$ is an unrolling of the transformation that approximately takes any $x_i$ to $\overline{x}_i$. Then Eq. (3) can be written as

$$\min_{\text{rank}(Z)=K} \sum_{i=1}^{n} \left\| \text{mat}(Z_i) x_i - (\overline{x}_i - x_i) \right\|_2, \tag{4}$$

where $\text{mat} : \mathbb{R}^{d^2} \to \mathbb{R}^{d \times d}$ is the matricization operator. Note the rank of $Z$ is at most $K$, the number of transformations. We then relax the rank constraint to a trace norm penalty as

$$\min \sum_{i=1}^{n} \left\| \text{mat}(Z_i) x_i - (\overline{x}_i - x_i) \right\|_2 + \lambda \left\| Z \right\|_*. \tag{5}$$

However, the matrix $Z \in \mathbb{R}^{n \times d^2}$ is too large to handle for real-world problems. Therefore, we propose approximating the objective function by modeling the transformation matrices as weighted sums of observed transformation pairs. This idea of using sampled pairs is similar to a kernel method: we will show that the true transformation matrices $A_k^*$ can be written as a linear combination of rank-one matrices $(\overline{x}_i - x_i)x_i^\top$. [1]

As intuition, assume that we are given a single point $x_i \in \mathbb{R}^d$ and $\overline{x}_i = t_i A^* x_i + x_i$, where $t_i \in \mathbb{R}$ is unobserved. If we approximate $A^*$ via the rank-one approximation $A = (\overline{x}_i - x_i)x_i^\top$, then $\|x_j\|_2^{-2} A x_i + x_i = \overline{x}_i$. This shows that $A$ captures the behavior of $A^*$ on a single point $x_i$. By sampling sufficiently many examples and appropriately weighting each example, we can construct an accurate approximation over all points.

Let us subsample $x_1, \ldots, x_r$ (WLOG, these are the first $r$ points). Given these samples, let us write a transformation $A$ as a weighted sum of $r$ rank-one matrices $(\overline{x}_j - x_j)x_j^\top$ with weights $\alpha \in \mathbb{R}^{n \times r}$. We then optimize these weights:

$$\min_\alpha \sum_{i=1}^n \left\| \sum_{j=1}^r \alpha_{ij}(\overline{x}_j - x_j)x_j^\top x_i - (\overline{x}_i - x_i) \right\|_2 + \lambda \|\alpha\|_* . \tag{6}$$

Next we show that with high probability, the weighted sum of $O(K^2 d)$ samples is close in operator norm to the true transformation matrix $A^*$ (Lemma 3.2 and Theorem 3.3).

## 3.2 Learning one transformation via subsampling

We begin by giving the intuition behind the sampling based objective in the one-transformation case. The correctness of rank-one reconstruction is obvious for the special case where the number of samples $r = d$, and for each $i$ we define $x_i = e_i$, where $e_i$ is the $i$-th canonical basis vector. In this case $\overline{x}_i = t_i A^* e_i + e_i$ for some unknown $t_i \in \mathbb{R}$. Thus we can easily reconstruct $A^*$ with a weighted combination of rank-one samples as $A = \sum_i A^* e_i e_i^\top = \sum_i \alpha_i (\overline{x}_i - x_i)x_i^\top$ with $\alpha_i = t_i^{-1}$.

In the general case, we observe the effects of $A^*$ on a non-orthogonal set of vectors $x_1 \ldots x_r$ as $\overline{x}_i - x_i = t_i A^* x_i$. A similar argument follows by changing our basis to make $t_i x_i$ the $i$-th canonical basis vector and reconstructing $A^*$ in this new basis. The change of basis matrix for this case is the map $\Sigma^{-1/2}$ where $\Sigma = \sum_{i=1}^r x_i x_i^\top / r$.

Our lemma below makes the intuition precise and shows that given $r > d$ samples, there exists weights $\alpha \in \mathbb{R}^d$ such that $A^* = \sum_i \alpha_i (\overline{x}_i - x_i)x_i^\top \Sigma^{-1}$, where $\Sigma$ is the inner product matrix from above. This justifies our objective in Eq. (6), since we can whiten $x$ to ensure $\Sigma = I$, and there exists weights $\alpha_{ij}$ which minimizes the objective by reconstructing $A^*$.

**Lemma 3.2.** *Given $x_1 \ldots x_r$ drawn i.i.d. from a density with full-rank covariance, and neighboring points $\overline{x}_i \ldots \overline{x}_r$ defined by $\overline{x}_i = t_i A^* x_i + x_i$ for some unknown $t_i \neq 0$ and $A^* \in \mathbb{R}^{d \times d}$.*

*If $r \geq d$, then there exists weights $\alpha \in \mathbb{R}^r$ which recover the unknown $A^*$ as*

$$A^* = \sum_{i=1}^r \alpha_i (\overline{x}_i - x_i)x_i^\top \Sigma^{-1},$$

*where $\alpha_i = 1/(rt_i)$ and $\Sigma = \sum_{i=1}^r x_i x_i^\top / r$.*

*Proof.* The identity $\overline{x}_i = t_i A^* x_i + x_i$ implies $t_i(\Sigma^{-1/2} A^* \Sigma^{1/2})\Sigma^{-1/2}x_i = \Sigma^{-1/2}(\overline{x}_i - x_i)$.

Summing both sides with weights $\alpha_i$ and multiplying by $x_i^\top (\Sigma^{-1/2})^\top$ yields

$$\sum_{i=1}^r \alpha_i \Sigma^{-1/2}(\overline{x}_i - x_i)x_i^\top (\Sigma^{-1/2})^\top = \sum_{i=1}^r \alpha_i t_i(\Sigma^{-1/2} A^* \Sigma^{1/2})\Sigma^{-1/2}x_i x_i^\top (\Sigma^{-1/2})^\top$$

$$= \Sigma^{-1/2} A^* \Sigma^{1/2} \sum_{i=1}^r \alpha_i t_i \Sigma^{-1/2}x_i x_i^\top (\Sigma^{-1/2})^\top.$$

By construction of $\Sigma^{-1/2}$ and $\alpha_i = 1/(t_i r)$, $\sum_{i=1}^r \alpha_i t_i \Sigma^{-1/2}x_i x_i^\top (\Sigma^{-1/2})^\top = I$. Therefore, $\sum_{i=1}^r \alpha_i \Sigma^{-1/2}(\overline{x}_i - x_i)x_i^\top (\Sigma^{-1/2})^\top = \Sigma^{-1/2} A^* \Sigma^{1/2}$. When $x$ spans $\mathbb{R}^d$, $\Sigma^{-1/2}$ is both invertible and symmetric giving the theorem statement. $\square$

## 3.3 Learning multiple transformations

In the case of multiple transformations, the definition of recovering any single transformation matrix $A_k^*$ is ambiguous since given transformations $A_1^*$ and $A_2^*$, the matrices $A_1^* + A_2^*$ and $A_1^* - A_2^*$ both locally generate the same family of transformations. We will refer to the transformations $A^* \in \mathbb{R}^{K \times d \times d}$ and strengths $t \in \mathbb{R}^{n \times K}$ as disentangled if $t^\top t / r = \sigma^2 I$ for a scalar $\sigma^2 > 0$. This criterion implies that the activation strengths are uncorrelated across the observed data. We will later

show in section 3.4 that this definition of disentangling captures our intuition, has a closed form estimate, and is closely connected to our optimization problem.

We show an analogous result to the one-transformation case (Lemma 3.2) which shows that given $r > K^2$ samples we can find weights $\alpha \in \mathbb{R}^{r \times k}$ which reconstruct any of the $K$ disentangled transformation matrices $A_k^*$ as $A_k^* \approx A_k = \sum_{i=1}^{r} \alpha_{ik}(\overline{x}_i - x_i)x_i^\top$.

This implies that minimization over $\alpha$ leads to estimates of $A^*$. In contrast to Lemma 3.2, the multiple transformation recovery guarantee is probabilistic and inexact. This is because each summand $(\overline{x}_i - x_i)x_i^\top$ contains effects from all $K$ transformations, and there is no weighting scheme which exactly isolates the effects of a single transformation $A_k^*$. Instead, we utilize the randomness in $t$ to estimate $A_k^*$ by approximately canceling the contributions from the $K - 1$ other transformations.

**Theorem 3.3.** *Let $x_1 \ldots x_r \in \mathbb{R}^d$ be i.i.d isotropic random variables and for each $k \in [K]$, define $t_{1,k} \ldots t_{r,k} \in \mathbb{R}$ as i.i.d draws from a symmetric random variable with $t^\top t / r = \sigma^2 I \in \mathbb{R}^{d \times d}$, $t_{ik} < C_1$, and $\|x_i\|^2 < C_2$ with probability one.*

*Given $x_1 \ldots x_r$, and neighbors $\overline{x}_1 \ldots \overline{x}_r$ defined as $\overline{x}_i = \sum_{k=1}^{K} t_{ik} A_k^* x_i + x_i$ for some $A_k^* \in \mathbb{R}^{d \times d}$, there exists $\alpha \in \mathbb{R}^{r \times K}$ such that for all $k \in [K]$,*

$$P\left(\left\|A_k^* - \sum_{i=1}^{r} \alpha_{ik}(\overline{x}_i - x_i)x_i^\top\right\| > \varepsilon\right) < Kd \exp\left(\frac{-r\varepsilon^2 \sup_k \|A_k^*\|^{-2}}{2K^2(2C_1^2 C_2^2(1 + K^{-1}\sup_k \|A_k^*\|^{-1}\varepsilon)}\right).$$

*Proof.* We give a proof sketch and defer the details to the supplement (Section 7). We claim that for any $k$, $\alpha_{ik} = \frac{t_{ik}}{\sigma^2 r}$ satisfies the theorem statement. Following the one-dimensional case, we can expand the outer product in terms of the transformation $A^*$ as

$$A_k = \sum_{i=1}^{r} \alpha_{ik}(\overline{x}_i - x_i)x_i^\top = \sum_{k'=1}^{K} A_{k'}^* \sum_{i=1}^{r} \alpha_{ik} t_{ik'} x_i x_i^\top.$$

As before, we must now control the inner terms $Z_{k'}^k = \sum_{i=1}^{r} \alpha_{ik} t_{ik'} x_i x_i^\top$. We want $Z_{k'}^k$ to be close to the identity when $k' = k$ and near zero when $k' \neq k$. Our choice of $\alpha_{ik} = \frac{t_{ik}}{\sigma^2 r}$ does this since if $k' \neq k$ then $\alpha_{ik} t_{ik'}$ are zero mean with random sign, resulting in Rademacher concentration bounds near zero, and if $k' = k$ then Bernstein bounds show that $Z_k^k \approx I$ since $\mathbb{E}[\alpha_{ik} t_i] = 1$. $\square$

### 3.4 Disentangling transformations

Given $K$ estimated transformations $A_1 \ldots A_K \in \mathbb{R}^{d \times d}$ and strengths $t \in \mathbb{R}^{n \times K}$, any invertible matrix $W \in \mathbb{R}^{K \times K}$ can be used to find an equivalent family of transformations $\hat{A}_i = \sum_k W_{ik} A_k$ and $\hat{t}_{ik} = \sum_j W_{kj}^{-1} t_{ij}$.

Despite this unidentifiability, there is a choice of $\hat{A}_1 \ldots \hat{A}_K$ and $\hat{t}$ which is equivalent to $A_1 \ldots A_K$ but *disentangled*, meaning that across the observed transformation pairs $\{(x_i, \overline{x}_i)\}_{i=1}^{n}$, the strengths for any two pairs of transformations are uncorrelated $\hat{t}^\top \hat{t} / n = I$. This is a necessary condition to captures the intuition that two disentangled transformations will have independent strength distributions. For example, given a set of images generated by changing lighting conditions and sharpness, we expect the sharpness of an image to be uncorrelated to lighting condition.

Formally, we will define a set of $\hat{A}$ such that: $\hat{t}_{.j}$ and $\hat{t}_{.i}$ are uncorrelated over the observed data, and any pair of transformations $\hat{A}_i x$ and $\hat{A}_j x$ generate decorrelated outputs. In contrast to mutual information based approaches to finding disentangled representations, our approach only seeks to control second moments, but enforces decorrelation both in the latent space ($t_{ik}$) as well as the observed space ($\hat{A}_i x$).

**Theorem 3.4.** *Given $A_k \in \mathbb{R}^{d \times d}$, $t \in \mathbb{R}^{n \times k}$ with $\sum_i t_{ik} = 0$, define $Z = USV^\top \in \mathbb{R}^{n \times d^2}$ as the SVD of $Z$, where each row is $Z_i = \sum_{k=1}^{K} t_{ik} vec(A_k)$.*

*The transformation $\hat{A}_k = S_{k,k} mat(V_k^\top)$ and strengths $\hat{t}_{ik} = U_{ik}$ fulfils the following properties:*

- $\sum_k \hat{t}_{ik} \hat{A}_k x_i = \sum_k t_{ik} A_k x_i$ *(correct behavior),*

- $\hat{t}^\top \hat{t} = I$ *(uncorrelated in latent space)*,

- $\mathbb{E}[\langle \hat{A}_i X, \hat{A}_j X \rangle] = 0$ *for any $i \neq j$ and random variable $X$ with $\mathbb{E}[XX^\top] = I$ (uncorrelated in observed space)*.

*Proof.* The first property follows since $Z$ is rank-$K$ by construction, and the rank-K SVD preserves $\sum_k t_{ik} A_k$ exactly. The second property follows from the SVD, $U^\top U = I$. The last property follows from $VV^\top = I$, implying $\mathrm{tr}(\hat{A}_i^\top \hat{A}_j) = 0$ for $i \neq j$. By linearity of trace: $\mathbb{E}[\langle \hat{A}_i X, \hat{A}_j X \rangle] = S_{i,i} S_{j,j} \, \mathrm{tr}(\mathrm{mat}(V_i) \mathrm{mat}(V_j)^\top) = 0$.

$\square$

Interestingly, this SVD appears in both the convex and subsampling algorithm (Eq. 6) as part of the proximal step for the trace norm optimization. Thus the rank sparsity induced by the trace norm naturally favors a small number of disentangled transformations.

## 4    Experiments

We evaluate the effectiveness of our sampling-based convex relaxation for learning transformations in two ways. In section 4.1, we check whether we can recover a known set of rotation / translation transformations applied to a downsampled celebrity face image dataset. Next, in section 4.2 we perform a qualitative evaluation of learning transformations over raw celebrity faces (CelebA) and MNIST digits, following recent evaluations of disentangling in adversarial networks [2].

### 4.1    Recovering known transformations

We validate our convex relaxation and sampling procedure by recovering synthetic data generated from known transformations, and compare these to existing approaches for learning linear transformations. Our experiment consists of recovering synthetic transformations applied to 50 image subsets of a downsampled version ($18 \times 18$) of CelebA. The resolution and dataset size restrictions were due to runtime restrictions from the baseline methods.

We compare two versions of our matrix Lie group learning algorithm against two baselines. For our method, we implement and compare convex relaxation with sampling (Eq. 6) and convex relaxation and sampling followed by gradient descent. This second method ensures that we achieve exactly the desired number of transformations $K$, since trace norm regularization cannot guarantee a fixed rank constraint. The full convex relaxation (Eq. 5) is not covered here, since it is too slow to run on even the smallest of our experiments.

As baselines, we compare to gradient descent with restarts on the nonconvex objective (Eq. 3) and the EM algorithm from Miao and Rao [11] run for 20 iterations and augmented with the SVD based disentangling method (Theorem 3.4). These two methods represent the two classes of existing approaches to estimating general linear transformations from pairwise data [11].

Optimization for our methods and gradient descent use minibatch proximal gradient descent with Adagrad [8], where the proximal step for trace norm penalties use subsampling down to five thousand points and randomized SVD. All learned transformations were disentangled using the SVD method unless otherwise noted (Theorem 3.4).

Figures 1a and b show the results of recovering a single horizontal translation transformation with error measured in operator norm. Convex relaxation plus gradient descent (Convex+Gradient) achieves the same low error across all sampled 50 image subsets. Without the gradient descent, convex relaxation alone does not achieve low error, since the trace norm penalty does not produce exactly rank-one results. Gradient descent on the other hand gets stuck in local minima even with stepsize tuning and restarts as indicated by the wide variance in error across runs. All methods outperform EM while using substantially less time.

Next, we test disentangling and multiple-transformation recovery for random rotations, horizontal translations, and vertical translations (Figure 1c). In this experiment, we apply the three types of transformations to the downsampled CelebA images, and evaluate the outputs by measuring the minimum-cost matching for the operator norm error between learned transformation matrices and

the ground truth. Minimizing this metric requires recovering the true transformations up to label permutation.

We find results consistent with the one-transform recovery case, where convex relaxation with gradient descent outperforms the baselines. We additionally find SVD based disentangling to be critical to recovering multiple transformations. We find that removing SVD from the nonconvex gradient descent baseline leads to substantially worse results (Figure 1c).

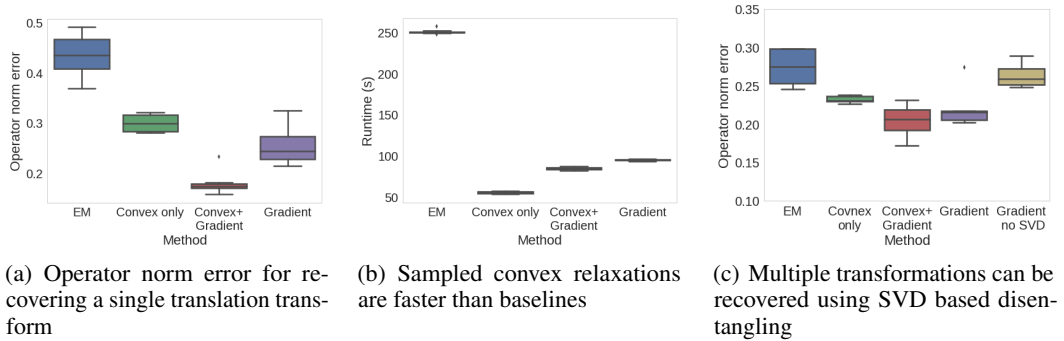

(a) Operator norm error for recovering a single translation transform

(b) Sampled convex relaxations are faster than baselines

(c) Multiple transformations can be recovered using SVD based disentangling

Figure 1: Sampled convex relaxation with gradient descent achieves lower error on recovering a single known transformation (panel a), runs faster than baselines (panel b) and recovers multiple disentangled transformations accurately (panel c).

## 4.2 Qualitative outputs

We now test convex relaxation with sampling on MNIST and celebrity faces. We show a subset of learned transformations here and include the full set in the supplemental Jupyter notebook.

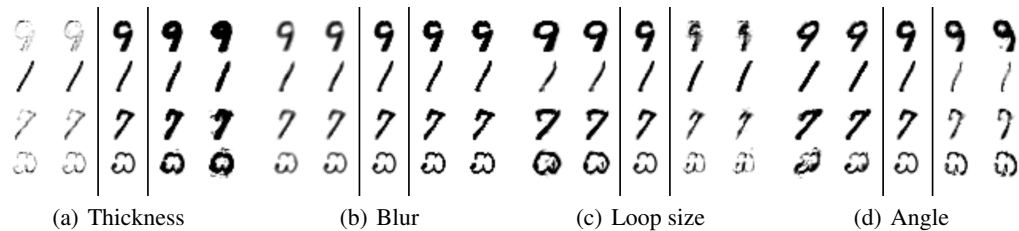

(a) Thickness      (b) Blur      (c) Loop size      (d) Angle

Figure 2: Matrix transformations learned on MNIST (top rows) and extrapolating on Kannada handwriting (bottom row). Center column is the original digit, flanking columns are generated by applying the transformation matrix.

On MNIST digits we trained a five-dimensional linear transformation model over a 20,000 example subset of the data, which took 10 minutes. The components extracted by our approach represent coherent stylistic features identified by earlier work using neural networks [2] such as thickness, rotation as well as some new transformations loop size and blur. Examples of images generated from these learned transformations are shown in figure 2. The center column is the original image and all other images are generated by repeatedly applying transformation matrices). We also found that the transformations could also sometimes extrapolate to other handwritten symbols, such as Kannada handwriting [5] (last row, figure 2). Finally, we visualize the learned transformations by summing the estimated transformation strength for each transformation across the minimum spanning tree on the observed data (See supplement section 9 for details). This visualization demonstrates that the learned representation of the data captures the style of the digit, such as thickness and loop size and ignores the digit identity. This is a highly desirable trait for the algorithm, as it means that we can extract continuous factors of variations such as digit thickness without explicitly specifying and removing cluster structure in the data (Figure 3).

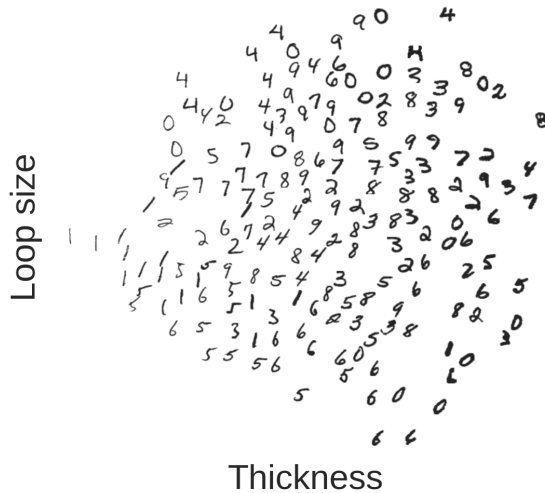

Loop size

Thickness

Figure 3: Embedding of MNIST digits based on two transformations: thickness and loop size. The learned transformations captures extracts continuous, stylistic features which apply across multiple clusters despite being given no cluster information.

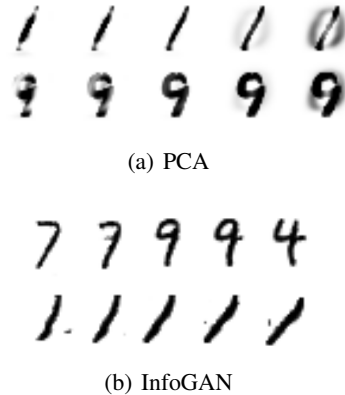

(a) PCA

(b) InfoGAN

Figure 4: Baselines applied to the same MNIST data often entangle digit identity and style.

In contrast to our method, many baseline methods inadvertently capture digit identity as part of the learned transformation. For example, the first component of PCA simply adds a zero to every image (Figure 4), while the first component of InfoGAN has higher fidelity in exchange for training instability, which often results in mixing digit identity and multiple transformations (Figure 4).

Finally, we apply our method to the celebrity faces dataset and find that we are able to extract high-level transformations using only linear models. We trained a our model on a 1000-dimensional PCA projection of CelebA constructed from the original 116412 dimensions with $K = 20$, and found both global scene transformation such as sharpness and contrast (Figure 5a) and more high level-transformations such as adding a smile (Figure 5b).

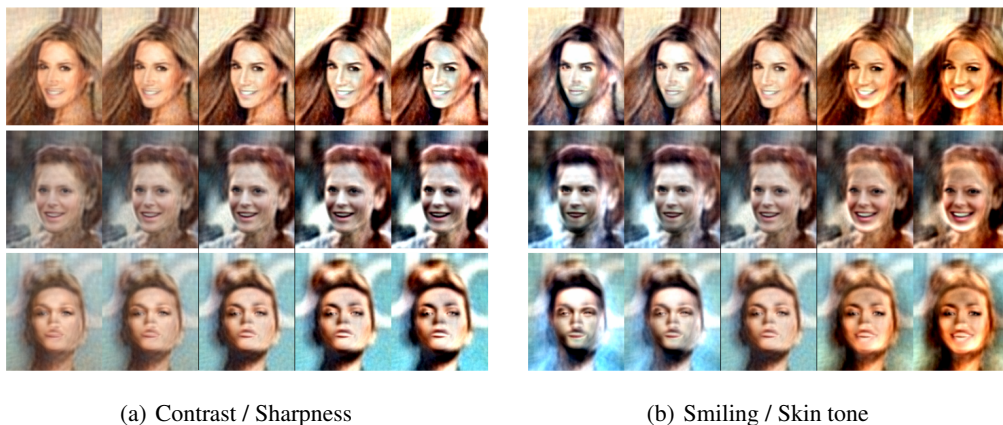

(a) Contrast / Sharpness

(b) Smiling / Skin tone

Figure 5: Learned transformations for celebrity faces capture both simple (sharpness) and high-level (smiling) transformations. For each panel, the center column is the original image, and columns to the left and right were generated by repeatedly applying the learnt transformation.

# 5   Related Work and Discussion

Learning transformation matrices, also known as Lie group learning, has a long history with the closest work to ours being Miao and Rao [11] and Rao and Ruderman [12]. These earlier methods use a Taylor approximation to learn a set of small ($< 10 \times 10$) transformation matrices given pairs of image patches undergoing a small transformation. In contrast, our work does not require supervision in the form of transformation pairs and provides a scalable new convex objective function.

There have been improvements to Rao and Ruderman [12] focusing on removing the Taylor approximation in order to learn transformations from distant examples: Cohen and Welling [3, 4] learned commutative and 3d-rotation Lie groups under a strong assumption of uniform density over rotations. Sohl-Dickstein et al. [14] learn commutative transformations generated by normal matrices using eigendecompositions and supervision in the form of successive $17 \times 17$ image patches in a video. Our work differs because we seek to learn multiple, general transformation matrices from large, high-dimensional datasets. Because of this difference, our algorithm focuses on scalability and avoiding local minima at the expense of utilizing a less accurate first-order Taylor approximation. This approximation is reasonable, since we fit our model to nearest neighbor pairs which are by definition close to each other. Empirically, we find that these approximations result in a scalable algorithm for unsupervised recovery of transformations.

Learning to transform between neighbors on a nonlinear manifold has been explored in Dollár et al. [7] and Bengio and Monperrus [1]. Both works model a manifold by predicting the linear neighborhoods around points using nonlinear functions (radial basis functions in Dollár et al. [7] and a one-layer neural net in Bengio and Monperrus [1]). In contrast to these methods, which begin with the goal of learning all manifolds, we focus on a class of linear transformations, and treat the general manifold problem as a special kernelization. This has three benefits: first, we avoid the high model complexity necessary for general manifold learning. Second, extrapolation beyond the training data occurs explicitly from the linear parametric form of our model (e.g., from digits to Kannada). Finally, linearity leads to a definition of disentangling based on correlations and a SVD based method for recovering disentangled representations.

In summary, we have presented an unsupervised approach for learning disentangled representations via linear Lie groups. We demonstrated that for image data, even a linear model is surprisingly effective at learning semantically meaningful transformations. Our results suggest that these semi-parametric transformation models are promising for identifying semantically meaningful low-dimensional continuous structures from high-dimensional real-world data.

**Acknowledgements.**

We thank Arun Chaganty for helpful discussions and comments. This work was supported by NSF-CAREER award 1553086, DARPA (Grant N66001-14-2-4055), and the DAIS ITA program (W911NF-16-3-0001).

**Reproducibility.**

Code, data, and experiments can be found on Codalab Worksheets (`http://bit.ly/2Aj5tti`).

## Footnotes

[1]Section 9 of the supplemental material introduces a kernelized version that extends this idea to general manifolds.
