[Supplementary Material]

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

# 6 Extra results

# 2-d visualization of MNIST

Figure 6: Visualization of MNIST in 2d coordinate system using the MST method, colored points indicate cluster identity, which was not used in training transformations.

Figure 7: Digit transformations learned by infogan across three runs. All transformations have examples which simultaneously rotate, thicken, and change digit identity

# 7 Proof of Theorem 3.3

Recall the proof statement:

**Theorem 7.1.** *Let $x$ be isotropic and $t_{ik}$ be symmetric random variables drawn independently of $x$ with $t^\top t/r = \sigma^2 I$, $t_{jk} < C_1$, and $\|x_j\|^2 < C_2$ with probability one.*

*Given $\overline{x}$ generated as*

$$\sum_{k=1}^{K} A_k t_{ik} x_i = \overline{x}_i, \tag{7}$$

*for every $k \in [K]$ there exists $\alpha_{jk} \in \mathbb{R}^r$ such that*

$$P\left(\left\|A_k - \sum_{j=1}^{r} \alpha_{jk}(\overline{x}_j - x_j)x_j^\top\right\| < \varepsilon\right) < K d \exp\left(\frac{-r\varepsilon^2 \sup_k \|A_k\|^{-2}}{2K^2(2C_1^2 C_2^2(1 + K^{-1} \sup_k \|A_k\|^{-1} \varepsilon)}\right).$$

*Proof.* For any $k$, $\alpha_{jk} = \frac{t_{jk}}{\sigma^2 r}$ fulfils the above property. Following the 1-D case we can expand the outer product in terms of the transformation $A$.

$$\sum_{j} \alpha_{jk}(\overline{x}_j - x_j)x_j^\top = \sum_{j=1}^{r}\sum_{k'=1}^{K} \alpha_{jk} t_{jk'} A_k x_j x_j^\top$$

$$= \sum_{k'=1}^{K} A_{k'} \sum_{j=1}^{r} \alpha_{jk} t_{jk'} x_j x_j^\top$$

The key quantity to control in this sum in order to estimate the transformation $A_k$ is the inner terms $Z_{k'}^k = \sum_{j=1}^{r} \alpha_{jk} t_{jk'} x_j x_j^\top$. We want $Z_{k'}^k$ to be close to the identity when $k' = k$ and near zero when $k' \neq k$.

**Case 1:** $k \neq k'$

In this case, $\sum_{j=1}^{r} \alpha_{jk} t_{jk'} = 0$ by construction of $\alpha$ and $t$ and we can write $Z_{k'}^k$ as a Rademacher sum to exploit this fact

$$Z_{k'}^k = \sum_{j=1}^{r} \mathrm{sign}(\alpha_{jk} t_{jk'})|\alpha_{jk} t_{jk'}| x_j x_j^\top.$$

This is a matrix Rademacher series and we can apply the standard bounds [15, Theorem 4.1.1] to obtain that
$$P(\|Z_{k'}^k\| > \varepsilon) \leq d \exp(-\varepsilon^2/(2v(Z_{k'}^k))).$$

the variance statistic is

$$v(Z_{k'}^k) = \left\|E\left[\sum_{j=1}^{r} |\alpha_{jk} t_{jk'}|^2 \|x_j\| x_j x_j^\top\right]\right\|.$$

Since $\|x_j\|^2 < C_1$ and $\mathbb{E}[\alpha_{jk}^2 t_{jk'}^2] \leq 1/r^2$ by independence of $t_{jk}$,

$$v(Z_{k'}^k) \leq \frac{C_1^2}{r}.$$

Thus the overall bound is

$$P(\|Z_{k'}^k\| > \varepsilon) \leq d \exp(-r\varepsilon^2/(2C_1^2)).$$

**Case 2:** $k = k'$

In this case, the construction of $\alpha$ and $t$ gives $\sum_{j=1}^{r} \alpha_{jk} t_{jk'} = \sum_{j=1}^{r} t_{jk}^2/r = 1$.

Using the fact that $t^\top t/r = \sigma^2 I$, the expected value of $Z_k^k$ is

$$E[Z_k^k] = E[t^2]E[xx^\top]/\sigma^2 = E[xx^\top] = I.$$

The Bernstein bound [15, Section 1.6.3] then implies that if $L \geq \|I - Z_k^k\|$ with probability one, then
$$P(\|I - Z_k^k\| > \varepsilon) \leq d \exp(-\varepsilon^2/(2v(Z_k^k + 2L\varepsilon/3)).$$

The variance statistic is defined as

$$v(Z_k^k) = \left\| \sum_{j=1}^r E\left[ (t_{jk}^2 x_j x_j^\top / r - I)^2 \right] \right\|.$$

Given $\|x_j\|^2 < C_1$ and $t_{jk'} < C_2$ the uniform upper bound $L$ is defined as

$$L = \left\| t_{jk}^2 x_j x_j^\top / r - I \right\| \le C_2^2 C_1 / r.$$

Which implies that the variance statistic is bounded above by

$$v(Z_k^k) \le \left\| \sum_{j=1}^r E\left[ (t_{jk}^2 x_j x_j^\top / r - I)^2 \right] \right\| \le C_1 C_2^2 / r.$$

Thus the overall bound is

$$P(\|I - Z_k^k\| > \varepsilon) \le d \exp(-r\varepsilon^2 / (2C_2^2 C_1 + 2C_2^2 C_1 \varepsilon/3)).$$

**Bounding the overall spectrum:** The overall error is bounded by the following

$$\left\| A_k - \sum_{k'=1}^K A_{k'} \sum_{j=1}^r \alpha_{jk} t_{jk'} x_j x_j^\top x \right\|$$

$$= \left\| A_k (I - Z_k^k) - \sum_{k \ne k} A_{k'} Z_{k'}^k \right\|$$

$$\le \left\| A_k (I - Z_k^k) \right\| + \left\| \sum_{k' \ne k} A_{k'} Z_{k'}^k \right\|$$

$$\le \|A_k\| \|(I - Z_k^k)\| + \sum_{k' \ne k} \|A_{k'}\| \|Z_{k'}^k\|$$

$$\le \sup_{k'} \|A_{k'}\| \left( \|(I - Z_k^k)\| + \sum_{k' \ne k} \|Z_{k'}^k\| \right).$$

Union bounding, and simplifying the bound,

$$P\left( \left\| A_k - \sum_{k'=1}^K A_{k'} \sum_{j=-1}^r \alpha_{jk} t_{jk'} \right\| > \varepsilon \right)$$

$$\le P\left( \|I - Z_k^k\| > \frac{\varepsilon}{K \sup_{k'} \|A_{k'}\|} \right)$$

$$+ \sum_{k' \ne k} P\left( \|Z_{k'}^k\| > \frac{\varepsilon}{K \sup_{k'} \|A_{k'}\|} \right)$$

$$< Kd \exp\left( \frac{-r\varepsilon^2}{2K^2 \sup_k \|A_k\|^2 \left( 2C_1^2 C_2^2 (1 + K^{-1} \sup_k \|A_k\|^{-1} \varepsilon) \right)} \right).$$

$\square$

## 8   Kernelization

We begin by defining the kernelization. Recall the one dimensional sampled transformation model given in terms of sampled $x$ and inner product matrix $\Sigma = x^\top x / r$,

$$\sum_{j=1}^r \alpha_j (\overline{x}_j - x_j) x_j^\top \Sigma^{-1} = A.$$

This formula resembles the multivariate ordinary least-squares estimator given predictors $x_j$ and responses $\alpha_j(\overline{x}_j - x_j)$. The equivalent dual form for kernel ridge regression with a kernel $\kappa$ (which we will later specialize to be Gaussian) gives the following kernelized matrix Lie group learning method of minimizing

$$\kappa_{\lambda_2,j}^{-1} = e_j^\top (\kappa(x,x) + \lambda_2 I)^{-1} \tag{8}$$

$$\min_\alpha \sum_{i=1}^n \left\| \sum_{j=1}^r \alpha_{ij}(\overline{x}_j - x_j)\kappa_{\lambda_2,j}^{-1}\kappa(x,x_i) - (\overline{x}_i - x_i) \right\|_2^2 \tag{9}$$

$$+ \lambda_1 \left\| \alpha \right\|_* . \tag{10}$$

Here, $\kappa(x,x)$ is the kernel matrix, $e_j$ is the $j$-th standard basis, and $\kappa(x,x_i) \in \mathbb{R}^n$ is the kernel vector between $x_i$ and the training data.

The kernelized problem in Equation 9 is not simply performing matrix Lie group learning after mapping to a Hilbert space, since the data $x$ is mapped via the kernel, but the pairwise differences $(\overline{x}_j - x_j)$ are in the original space. Equation 9 instead represents a multi-task kernel ridge regression problem of predicting $(\overline{x}_j - x_j)$.

We will now show that low-rank kernel ridge regression can model any $K$-dimensional parallelizable manifold $M$. A $K$-dimensional parallelizable manifold is a differentiable manifold equipped with $K$ smooth, complete vector fields such that the vector field spans the tangent space of $M$ at every point. These vector fields should be thought of both as transformations and a coordinate system.

**Theorem 8.1.** *Let $M$ be a parallelizable manifold of dimension $K$ with a compactly supported density $p$ on $M$ bounded strictly above zero, and $F_1 \ldots F_k$ be the smooth, unit length vector fields associated with $M$.*

*Let $x_{jk} \in \mathbb{R}^{Kr \times d}$ be $r$ draws from $p$, repeated $K$ times each (such that $x_{11} = x_{12} \ldots x_{1K}$) and $\overline{x}_{1k}$ be the $k$-th nearest neighbor of $x_{1k}$ over the $r$ draws.*

*Then there exists a parameter matrix $V^* \in \mathbb{R}^{K \times Kr}$ such that for $\kappa(x,y) = \exp(-\left\|x-y\right\|_2 /\sigma)$ and $\kappa_{\lambda_2}^{-1} = (\kappa(x,x) + \lambda_2 I)^{-1}$, the weighted kernel estimate of*

$$f_k(x'|x) = \sum_{j=1}^r \sum_{l=1}^K V^*_{klj}(\overline{x}_{jl} - x_{jl})\kappa_{\lambda_2 j}^{-1}\kappa(x,x')$$

*has the following expected held out error for any $k \in [K]$*

$$E_{x,x'\sim p}\left[\left\| f_k(x'|x) - F_k(x') \right\|_2\right] = O\left( \left( \frac{\log^2 r}{r} \right)^{1/(8+4K)} + \sqrt{K/\lambda_2}\left( \frac{K}{r} \right)^{3/K} \right)$$

*for appropriate choice of $\sigma, \lambda_2$.*

The proof of learning parallelizable manifolds mirrors our argument for the matrix Lie group case. First we show that there exists a rank $K$ weight matrix $\alpha$ which can be used to map the observed vector differences $(\overline{x}_i - x_i)$ into the tangent space of the manifold. Next, we show that under the appropriate neighborhood construction scheme, the kernel ridge regression predictor decomposes.

The supporting lemma on local coordinate parametrization is below:

**Lemma 8.2.** *Let $M$ be a parallelizable $K$-dimensional manifold with some compact density $p$ strictly bounded away from zero , and $F_1 \ldots F_k$ be the smooth, unit length vector fields associated with $M$.*

*Let $x_{jk} \in \mathbb{R}^{Kr \times d}$ be $r$ draws from $p$, repeated $K$ times each (such that $x_{11} = x_{12} \ldots x_{1K}$) and $\overline{x}_{1k}$ is the $k$-th nearest neighbor of $x_{1k}$ among the $r$ draws.*

*Then there exists a matrix $V \in \mathbb{R}^{r \times K \times K}$ such that for all $i \in 1 \ldots r$,*

$$\sum_{k=1}^K \left\| \sum_{l=1}^K ((\overline{x}_{il} - x_{il})V_{ilk}) - F_k(x_{i1}) \right\|_2 = O\left( \sqrt{K}\left( \frac{K}{r} \right)^{1/K} \right)$$

*Proof.* Since $x$ lie on a parallelizable manifold, for each $x_{il}$ there exists some $t_{il1} \ldots t_{ilK}$ and $F_k$ such that:

$$\overline{x}_{il} = \exp\left(\sum_{k=1}^{K} t_{ilk} F_k(x_{il})\right) x_{il}.$$

The $\exp$ here is the *exponential map* of the manifold $M$, which can be interpreted as following the geodesic starting at $x_{il}$ with initial velocity $\sum_{k=1}^{K} t_{ilk} F_k(x_{il})$ for unit time.

Define the linearization $\tilde{F}_k(x) = x + \sum_{k=1}^{K} t_{ilk} F_k(x)$. The smoothness of the manifold $M$ provides bounds on the accuracy of the linearization for all $x$ over small values for $t_{ilk}$ [10, proposition 20.10]:

$$\sum_{k=1}^{K} \left\|\tilde{F}_k(x_{il}) - \overline{x}_{il}\right\|_2 = O\left(\sum_{k=1}^{K} t_{ilk}^2\right)$$

Next, note that the linearization can be re-written in matrix form as $\tilde{F}_k(x_{il}) = x_{il} + \hat{F}_i T_i$ using

$$\hat{F}_i = \left[\begin{array}{c|c|c|c} F_1(x_{i1}) & F_2(x_{i1}) & \ldots & F_K(x_{i1}) \end{array}\right].$$

$$T_i = \left[\begin{array}{c|c|c|c} t_{i1} & t_{i2} & \ldots & t_{iK} \end{array}\right].$$

Also define the analogous column-wise stacked representation for the difference of neighbors:

$$\Delta \hat{X}_i = \left[\begin{array}{c|c|c|c} \overline{x}_{i1} - x_{i1} & \overline{x}_{i2} - x_{i2} & \ldots & \overline{x}_{iK} - x_{iK} \end{array}\right].$$

In this notation, the previous statement on linearization corresponds to:

$$||\Delta \hat{X}_i - \hat{F}_i T_i||_{2,1} \le O\left(\sum_{k=1}^{K} t_{kil}^2\right)$$

and we can apply a operator norm bound to obtain:

$$||\Delta \hat{X}_i T_i^{-1} - \hat{F}_i||_{2,1} \le ||T_i^{-1}||_2 O\left(\sum_{k=1}^{K} t_{kil}^2\right)$$

Finally, note that $||T_i^{-1}||_2 \le \sqrt{K}||T_i^{-1}||_1 = O(\sqrt{K}\left(\frac{K}{r}\right)^{-1/K})$, where the final equality follows from the construction of $\overline{x}$ as a nearest neighbor and the decay rate of nearest neighbor distance as $O\left(\frac{K}{r}\right)^{1/K}$ [6].

Finally, the same estimate of $O(|t_{ilk}|) = (K/r)^{1/K}$ gives the overall bound with $V_{ilk} = T_{ilk}$. $\square$

Next, we show a simple identity for kernel ridge regressions with repeated inputs:

**Lemma 8.3.** *Consider input $x_1 \ldots x_m$ such that $x_1 = x_2 = \ldots x_k$ and let $e_j$ be the $j$-th standard basis vector.*

*Then the ridge regression function defined by*

$$h_j(x') = e_j \left(\kappa(x, x) + \lambda_2 I\right)^{-1} \kappa(x, x')$$

*and*

$$h(x') = \sum_j y_j \alpha_j h_j(x')$$

*is equivalent to*

$$h(x') = \left(\sum_{j=1}^{k} y_j \alpha_j\right) h_1(x') + \sum_{j=k+1}^{m} y_j \alpha_j h_j(x').$$

*Proof.* Write the kernel matrix in block form with $k \times k$ and $(m-k) \times (m-k)$ sized blocks as

$$\kappa(x,x) = \left[ \begin{array}{c|c} \kappa_{11} & \kappa_{12} \\ \hline \kappa_{12}^\top & \kappa_{22} \end{array} \right].$$

By the block inversion formula we have

$$(\kappa(x,x) + \lambda_2 I)^{-1} = \left[ \begin{array}{c|c} A_1^{-1} & -(\kappa_{11} - \lambda_2 I)^{-1} \kappa_{12} A_2^{-1} \\ \hline -A_2^{-1} \kappa_{12}^\top (\kappa_{11} + \lambda_2 I)^{-1} & A_2^{-1}. \end{array} \right]$$

The upper and lower diagonal blocks are

$$A_1 = \kappa_{11} + \lambda_2 I - \kappa_{12}(\kappa_{22} + \lambda_2 I)^{-1} \kappa_{12}^\top$$
$$A_2 = \kappa_{22} + \lambda_2 I - \kappa_{12}^\top (\kappa_{11} + \lambda_2 I)^{-1} \kappa_{12}.$$

First we show that $A_1^{-1}$ can be written as the sum of a constant matrix and weighted identity matrix. Since the inputs $x_1 \ldots x_K$ are identical over the top blocks, $\kappa_{12}$ can be written $\kappa_{12} = 1 V_{12}^\top$ for some vector $V_{12}$ and

$$C_1 = c 1 1^\top + \lambda_2 I - 1 V_{12}^\top (\kappa_{22} + \lambda_2 I)^{-1} V_{12} 1^\top.$$

Thus $C_1$ is the sum of the identity scaled by $\lambda_2$, and the all ones matrix scaled by $c - V_{12}^\top (\kappa_{22} + \lambda_2 I)^{-1} V_{12}$. The Woodbury inversion lemma shows that the inverse is also a scaled identity plus a scaled all-ones matrix.

Second, we show that the top right block $-(\kappa_{11} + \lambda_2 I)^{-1} \kappa_{12} A_2^{-1}$ has identical rows. Applying the same argument,

$$- (\kappa_{11} + \lambda_2 I)^{-1} \kappa_{12} A_2^{-1}$$
$$= -\lambda_2^{-1}(I - c/(\lambda_2 + c) 1 1^\top) 1 v_{12}^\top C_2^{-1}$$
$$= 1 \left[ -\lambda_2^{-1} - c/(\lambda_2 + c) d \right] V_{12}^\top C_2^{-1}.$$

For shorthand, let $C_1^{-1} = c_1 I + c_2 1 1^\top$ and $-(\kappa_{11} + \lambda_2 I)^{-1} \kappa_{12} A_2^{-1} = 1 U_{12}^\top$ for some constants $c_1, c_2$ and vector $U_{12}$. The above two arguments show we can always find such $c_1, c_2, U_{12}$.

Finally, note that for all $i \in 1 \ldots k$,

$$h_i(x') = e_i \left( \kappa(x,x) \lambda_2 \right) \kappa(x,x')$$
$$= \left( \sum_{j=1}^K \kappa(x,x')_j c_2 \right) + \kappa(x,x')_i (c_1 - c_2) + \kappa(x,x')^\top U_{12}$$
$$= \kappa(x,x')_1 (c_1 + (K-1) c_2) + \kappa(x,x')^\top U_{12}.$$

The last equality uses the fact that $\kappa(x,x')_1 = \kappa(x,x')_2 \ldots = \kappa(x,x')_k$ since blocks of $k$ entries of $x$ are identical.

Therefore $h_1(x) = h_2(x) \ldots h_k(x)$, and we can conclude our original statement that

$$h(x) = \sum_j y_j \alpha_j h_j(x)$$

can be written as

$$h(x) = \left( \sum_{j=1}^k y_j \alpha_j \right) h_1(x) + \sum_{j=k+1}^m y_j \alpha_j h_j(x).$$

$\square$

Finally, the proof of the original theorem follows directly from standard kernel ridge regression bounds.

*Proof.* First, lemma 8.2 guarantees the existence of some $V_{ilk}$ such that for all $k \in [K]$,

$$\left\| \sum_{l=1}^{K} (\overline{x}_{il} - x_{il}) V_{ilk} - F_k(x) \right\|_2 = O\left( \sqrt{K} \left( \frac{K}{r} \right)^{1/K} \right).$$

So for each regression $k = 1 \ldots K$, set $V_{kli}^* = V_{ilk} \in \mathbb{R}^{K \times Kr}$.

Applying lemma 8.3 and collapsing the sums over the $k$ identical elements in each group $x_{i1} = \ldots = x_{iK}$ shows that the weighted kernel estimate is

$$f_k(x'|x) = \sum_{j=1}^{r} \left[ \sum_{l=1}^{K} V_{kli}(\overline{x}_{jl} - x_{jl}) \right] (\kappa(x, x) + \lambda_2 I)_j^{-1} \kappa(x, x'),$$

with $\kappa(x, x)_{ij} = \kappa(x_{i1}, x_{j1})$.

Now define the kernel ridge regression on the true transformation as

$$\hat{f}_k(x'|x) = \sum_{j=1}^{r} F_k(x_{j1})(\kappa(x, x) + \lambda_2 I)_j^{-1} \kappa(x, x').$$

Since $F_k$ is Lipschitz continuous by definition, choosing $\lambda = \left( \frac{\log^2(r)}{r} \right)^{(1+K)/(8+4K)}$ and any $\sigma > 0$,

$$E_{x, x' \sim p} \left[ \left\| \hat{f}_k(x'|x) - F_k(x') \right\|_2 \right] = O\left( \left( \frac{\log^2 r}{r} \right)^{1/(8+4K)} \right)$$

from the result of [17].

Using $\left\| \sum_{l=1}^{K} (\overline{x}_{il} - x_{il}) V_{ilk} - F_k(x) \right\|_2 = O\left( \sqrt{K} \left( \frac{K}{r} \right)^{1/K} \right)$ we have the overall bound of

$$E_{x, x' \sim p} \left[ \left\| f_k(x'|x) - F_k(x') \right\|_2 \right] = O\left( \left( \frac{\log^2 r}{r} \right)^{1/(8+4K)} + \sqrt{K/\lambda_2} \left( \frac{K}{r} \right)^{1/K} \right)$$

$\square$

# 9 Algorithm for embeddings based on transformations

Below is the detailed algorithm for taking a set of disentangled transformation matrices and generating a low-dimensional representation where each point is defined as being the image of repeatedly applying a transformation starting at some origin point $v_0$.

The algorithm forms a minimum spanning tree over data points and integrates the activations $t$ which are estimated by projecting the difference $(x_{v_i} - x_{h_i})$ between points spanning an edge of the MST to the span of the transformations $A_1 \ldots A_k$ at $x_{v_i}$ using the pseudoinverse.

---
**Algorithm 1** Representations from matrix Lie groups

---
**Require:** Data $x_1 \ldots x_n$, transformations $A_1 \ldots A_K$
1: Construct a MST $M$ over $x_1 \ldots x_n \in \mathbb{R}^d$.
2: Let $v_0, v_1 \ldots v_{n-1}$ be a breadth-first traversal of the tree starting at any root vertex $v$, and $h_0, h_1 \ldots h_{n-1}$ be the predecessor list such that $(v_i, h_i)$ is a MST edge.
3: Set $\overline{x}_v = 0$
4: **for** each $i \in 1 \ldots n$ **do**
5:

$$\overline{x}_{v_i} = \overline{x}_{h_i} + \left[ A_1 x_{v_i} \|A_1 x_{v_i}\|_2^{-1} \middle| \ldots \middle| A_k x_{v_i} \|A_k x_{v_i}\|_2^{-1} \right]^+ (x_{v_i} - x_{h_i})$$

6: **end for**
7: **return** $\overline{x}$.

---

# 10   Correctness of sampling the trace norm

We show that computing the trace norm over small subsets of $L$ rows of the parameter matrix is a reasonable approximation to the full trace norm:

**Theorem 10.1.** *Let $X$ be a $n \times r$ matrix with rank $K$, $Y$ be the $L \times r$ matrix formed by sampling $L$ rows of $X$ with replacement, and $||X||_* = tr(\sqrt{X^T X})$ the trace norm.*

*Assuming that $||X_i|| \leq B$ and the minimum nonzero eigenvalue of $X^\top X/r \geq \sigma_{\min}^2$,*

$$P\left( \left| ||Y||_*/\sqrt{L} - ||X||_*/\sqrt{n} \right| > \frac{K\varepsilon}{\sigma_{\min} + \sqrt{\sigma_{\min}^2 - \varepsilon}} \right) \leq 2r \exp\left( \frac{-L\varepsilon^2/2}{B||X^T X/r|| + 2B\varepsilon/3} \right)$$

*for all $\varepsilon < \sigma_{\min}^2$.*

*Proof.* First, by standard Bernstein high probability bounds in Tropp [15, Section 1.6.3] we have that

$$P(||Y^T Y/L - X^T X/r|| > \varepsilon) \leq 2r \exp\left( \frac{-L\varepsilon^2/2}{B||X^T X/r|| + 2B\varepsilon/3} \right).$$

Let $W_K$ be the orthogonal projection of $X$ onto its K-dimensional image. For convience define $\overline{X} = XW_k$ and $\overline{Y} = YW_K$. Since $W_k$ is an orthogonal projection onto the image of $X$ it preserves the singular values as $\sigma_i(X) = \sigma_i(\overline{X})$ and $\sigma_i(Y) = \sigma_i(\overline{Y})$ for all $i \in [K]$.

Next, by the Ando-Hemmen inequality [16],

$$\left\| \left( \overline{Y}^\top \overline{Y}/L \right)^{1/2} - \left( \overline{X}^\top \overline{X}/r \right)^{1/2} \right\| \leq \frac{||\overline{Y}^T \overline{Y}/L - \overline{X}^T \overline{X}/r||}{\sigma_{\min}(\overline{Y}) + \sigma_{\min}(\overline{X})}.$$

If $||\overline{Y}^T \overline{Y}/L - \overline{X}^T \overline{X}/r|| < \varepsilon$ then

$$\max_i |\sigma_i(Y) - \sigma_i(X)| \leq \left\| \left( \overline{Y}^\top \overline{Y}/L \right)^{1/2} - \left( \overline{X}^\top \overline{X}/r \right)^{1/2} \right\| \leq \frac{\varepsilon}{\sigma_{\min}(X) + \sqrt{\sigma_{\min}(X)^2 - \varepsilon}}.$$

Summing over all $K$ singular values,

$$P\left( \left\| \left| \sum_{i=1}^K \sigma_i\left( \frac{Y}{\sqrt{L}} \right) \right| - \left| \sum_{i=1}^K \sigma_i\left( \frac{X}{\sqrt{r}} \right) \right| \right\| > \frac{K\varepsilon}{\sigma_{\min}(X) + \sqrt{\sigma_{\min}(X)^2 - \varepsilon}} \right)$$
$$\leq 2r \exp\left( \frac{-L\varepsilon^2/2}{B||X^T X/r|| + 2B\varepsilon/3} \right).$$

$\square$