[Reviews · NeurIPS 2017]

Reviewer 1



The paper proposes an algorithm for unsupervised learning of transformations based on modeling nearest neighbor pairs as linear combination of transforms. The technique only models the transformations, and not the full data distribution and so can (in principle) be applied to other data sets (for eg, learning from MNIST, but applying to characters from other languages). While this problem/objective function is non-convex, they provide a convex relation by approximating the true transformation matrix as a linear combination of rank-1 matrices derived from sampling the data (and they show that this is a good approximation to the true transform matrix). They show that the technique recovers known transforms such as stroke thickness/rotation in addition to new transforms (blur, loop size). Overall, I found the ideas in the paper interesting, and the paper well written.